# The *Helicobacter pylori* HspR-Modulator CbpA Is a Multifunctional Heat-Shock Protein

**DOI:** 10.3390/microorganisms8020251

**Published:** 2020-02-13

**Authors:** Simona Pepe, Vincenzo Scarlato, Davide Roncarati

**Affiliations:** Department of Pharmacy and Biotechnology (FaBiT), University of Bologna, 40126 Bologna, Italy; simona.pepe4@unibo.it

**Keywords:** heat-shock response, CbpA, *Helicobacter pylori*, DNA binding activity, chaperones

## Abstract

The medically important human pathogen *Helicobacter pylori* relies on a collection of highly conserved heat-shock and chaperone proteins to preserve the integrity of cellular polypeptides and to control their homeostasis in response to external stress and changing environmental conditions. Among this set of chaperones, the CbpA protein has been shown to play a regulatory role in heat-shock gene regulation by directly interacting with the master stress-responsive repressor HspR. Apart from this regulatory role, little is known so far about CbpA functional activities. Using biochemistry and molecular biology approaches, we have started the in vitro functional characterization of *H. pylori* CbpA. Specifically, we show that CbpA is a multifunctional protein, being able to bind DNA and to stimulate the ATPase activity of the major chaperone DnaK. In addition, we report a preliminary observation suggesting that CbpA DNA-binding activity can be affected by the direct interaction with the heat-shock master repressor HspR, supporting the hypothesis of a reciprocal crosstalk between these two proteins. Thus, our work defines novel functions for *H. pylori* CbpA and stimulates further studies aimed at the comprehension of the complex regulatory interplay among chaperones and heat-shock transcriptional regulators.

## 1. Introduction

All living organisms respond to changing external conditions by synthetizing a class of highly conserved proteins whose function is to assist protein folding and to get rid of dangerous cytoplasmic aggregates made of denatured polypeptides. Transcriptional and posttranscriptional regulatory mechanisms provide a rapid and prompt accumulation of heat-shock proteins only when it is necessary, thereby ensuring an efficient expression of these protective effectors [1]. The pervasive gastric pathogen *Helicobacter pylori* is no exception to this general rule. It possesses almost all the members of the typical heat-shock proteins collection, which includes the GroEL/GroES chaperonin and the DnaK/DnaJ/GrpE chaperone system. The expression of these core players of *H. pylori* heat-shock response is tightly regulated by the concerted action of transcriptional and post-transcriptional mechanisms [2]. Specifically, the two widespread repressors HspR and HrcA negatively control transcription of the three multicistronic operons containing the major chaperones’ encoding genes. In addition, our group has previously demonstrated that chaperones themselves are able to exert a protein–protein mediated post-transcriptional regulation of the two heat-shock repressors. The chaperonin GroE directly interacts with the HrcA repressor and enhances its DNA binding activity, thereby playing a positive feedback role in the transcriptional control of chaperone genes [3]. We have also shown that the protein encoded by HPG27_RS02130, the first gene of the *hspR*-containing operon, named CbpA, negatively affects the regulatory activity of the heat-shock repressor. CbpA, in fact, interacts with HspR and hinders its binding to target promoters in vitro only when the repressor is not bound to its target sites, and without directly contacting the DNA. In addition, overexpression of CbpA leads to deregulation of heat-shock response in vivo, supporting the idea that this protein plays a regulatory role in *H. pylori* heat-shock response [4].

Since *H. pylori* complete genome sequencing and gene annotation, the protein product encoded by the gene HPG27_02130 has been referred to as CbpA, because of its primary sequence similarity with the better characterized CbpA protein of the model organism *Escherichia coli* [5]. In this latter bacterium, this protein was originally isolated for its ability to bind to AT-rich curved DNA in vitro and named ‘‘Curved DNA binding protein A′′ [6]. Later on, it has been shown that CbpA is a multifunctional protein, having both co-chaperone and DNA binding activities. CbpA, in fact, constitutes a functional homolog of DnaJ and its overexpression can complement all known phenotypes associated with DnaJ inactivation [7,8,9]. In addition, it has been demonstrated that CbpA accumulates in the stationary phase of bacterial growth and, following interaction with DNA and the formation of protein–DNA aggregates, it is able to protect nucleic acids from damage, acting as a nucleoid-associated protein [10]. The detailed mechanism of DNA binding has been dissected recently and shown to involve the recognition of the DNA minor groove [11]. Of note is the observation that both co-chaperone and DNA-binding activities of CbpA are regulated through direct interaction with a specific partner protein called CbpM (for CbpA modulator). This protein regulator is encoded by the gene laying immediately downstream *cbpA* within the same operon. The CbpA regulator CbpM essentially works as an inhibitor, forming a dimer able to bind two copies of the CbpA J-domain [12,13].

In this study, we sought to better understand *H. pylori* CbpA functional activity. Starting from a primary sequence comparison between *H. pylori* and *E. coli* CbpA proteins, we show that *H. pylori* CbpA has a co-chaperone activity, being able to stimulate DnaK ATPase activity in vitro. In addition, we demonstrate that *H. pylori* CbpA is a dimeric DNA-binding protein, and that CbpA dimerization is necessary for DNA recognition and binding. Interestingly, we also report preliminary evidences supporting the possibility that the heat-shock repressor HspR could negatively modulate DNA-binding activity of CbpA. This latter observation stimulates further studies to better characterize the functional relationship between HspR and CbpA, especially in the direction of defining the intersection between heat-shock genes transcriptional regulation and CbpA roles in the medically important human pathogen *H. pylori*.

## 2. Materials and Methods 

### 2.1. Molecular Biology Procedures

Common molecular biology techniques including plasmid DNA extraction, transformation, restriction, cloning, gel electrophoresis, and purification of DNA fragments were performed according to standard molecular biology techniques as described previously [14]. All restriction and modification enzymes were used according to the manufacturers’ instructions (New England Biolabs, Ipswich, MA, USA). All oligonucleotides used in this study have been designed based on the *H. pylori* G27 RefSeq annotation (GCF_000021165.1) in the version released in September 2017. For the construction of the plasmid expressing the C-terminal deleted mutant of CbpA (CbpA-ΔCt), the protein coding sequence from amino acid 1 to 276 was PCR amplified from *H. pylori* G27 genomic DNA with primers cbpAN (ATATcatatgAGCAAGAGTTTATACCAAAC) and cbpAD12R (ATATctcgagAAGCGTTTCGGTTTTAGGCAAAATCAAACG). Then, the polymerase chain reaction (PCR) amplicon was digested and cloned into NdeI/XhoI sites of the pET22b vector (Merck, Darmstadt, Germany), generating the final expression construct pET22b-CbpA-ΔCt.

### 2.2. Protein Expression and Purification

Carboxyl-terminal His-tagged *H. pylori* CbpA (CbpA-wt (wild type) and CbpA-ΔCt), DnaK, DnaJ, GrpE and HspR proteins were overexpressed in *Escherichia coli* BL21 (DE3) cells and affinity purified as previously described [15,16]. Following purification, the recombinant proteins were dialyzed against two changes of 1× binding buffer (10 mM Tris-HCl, pH 8.0; 50 mM NaCl; 10 mM KCl; 5 mM MgCl_2_; 0.1 mM DTT; 0.01% NP40), aliquoted and stored at −80 °C. Protein concentration was determined by Bradford colorimetric assay (BioRad, Hercules, CA, USA) and purity of the protein preparations was analyzed by denaturing sodium dodecyl sulfate–polyacrylamide gel electrophoresis (SDS-PAGE).

### 2.3. Colorimetric Determination of ATPase Activity

The ATPase assay protocol was adopted from previous reports [17,18]. Fresh solutions of Quinaldine Red (0.05% *w*/*v*), polyvinyl alcohol (2% *w*/*v*), ammonium heptamolybdate tetrahydrate (6% *w*/*v* in 6 M HCl) were prepared and mixed with water in the ratio of 2:1:1:2 to prepare the Quinaldine Red reagent (QR reagent). The different combinations of chaperone proteins were prepared in a final volume of 10 μL of binding buffer in the following ratios: DnaK:GrpE (1.0:1.5 μM), DnaK:GrpE:DnaJ (1.0:1.5:2.0 μM), DnaK:GrpE:CbpA (1.0:1.5:2.0 μM). The volume of each protein mixture was brought to 100 μL with binding buffer without glycerol and incubated at 37 °C for 30 minutes. Then, aliquots of each mixture (18 μL) were transferred to new tubes and 2 μL of 10 mM ATP (or 2 μL of H_2_O as negative control) were added, before incubating the reactions at 37 °C for 3 hours. After that, 80 μL of QR reagent were added to each tube and samples were incubated for 3 minutes at 37 °C. The reactions were quenched by adding 8 μL of 34% sodium citrate and incubated for 15 minutes at room temperature. The optical density of each sample at 528 nm was determined by means of a spectrophotometer. To control these results for intrinsic hydrolysis, the signal from ATP in identically treated buffer lacking chaperones was subtracted. To permit comparisons between screens performed at different times, a phosphate standard curve (using sodium phosphate) was generated each day.

### 2.4. In Vitro DNA Binding Assay

In vitro DNA binding studies were carried out essentially as previously described for the *H. pylori* HU nucleoid associated protein [19]. In detail, the purified CbpA-wt and CbpA-ΔCt proteins were incubated with a DNA probe consisting of the pHel2-flaAfur plasmid (a plasmid DNA deriving from a cryptic plasmid isolated from *H. pylori* [20]) and separated through electrophoresis on a native agarose gel. In detail, 25 ng of the plasmid DNA probe were mixed with increasing concentrations (specified in the figure legends) of CbpA-wt or CbpA-ΔCt purified proteins in 20 μL of 1× binding buffer. Following a 25 minutes incubation at 25 °C, the binding reactions were loaded on a 0.9% (*w*/*v*) agarose, 1× Tris-Borate-EDTA gel and run at 80 V for 120 minutes. After electrophoretic separation, DNA bands were visualized with ethidium bromide staining.

### 2.5. Chemical Crosslinking of Proteins

In vitro cross-linking assay was carried out by incubating 5 μM of the purified CbpA-wt and CbpA-ΔCt proteins with increasing concentrations (0.0000%, 0.0100%, 0.0050%, 0.0025%) of glutaraldehyde (Sigma-Aldrich, St Louis, MO, USA) in 15 μL of 1× binding buffer for 1 hour at 25 °C. Chemical crosslinking was halted by adding 5 μL of 5× SDS-PAGE loading buffer [14] and boiling each sample at 100 °C for 5 minutes. Then, reactions were separated by SDS-PAGE and stained with Coomassie Brilliant Blue R-250 (Sigma-Aldrich, USA).

## 3. Results

### 3.1. Sequence Similarity between Helicobacter pylori (H. pylori) and Escherichia coli (E. coli) CbpA

The *H. pylori* HPG27_RS02130 gene is a member of a three-cistronic operon that includes the genes encoding a transcriptional repressor (HspR) and a putative helicase (RarA). Since the complete genome sequencing of the *H. pylori* G27 strain, this gene has been annotated as *cbpA* and *dnaJ1*, for its deduced amino-acid sequence similarity with the well-characterized CbpA protein of *E. coli*. This latter protein consists of three functional domains: an N-terminal J domain (highlighted in green in Figure 1) which confers co-chaperone activity to the protein, a C-terminal domain divided into two subdomains (named CTDI and CTDII, highlighted in Figure 1 in cyan and red, respectively) responsible for DNA-binding (CTDI) and dimerization (CTDII), and a flexible linker (aa 76 to 120, not highlighted in Figure 1), partially involved in DNA binding [8].

Before starting a functional characterization of *H. pylori* CbpA, we decided to examine the primary sequence conservation between *H. pylori* and *E. coli* CbpA more in detail. Overall, *H. pylori* CbpA shows a high degree of similarity (33% identity) to the *E. coli* CbpA. Inspection of the primary structure of both proteins indicated that the highest conservation (45% identity) between the two sequences is confined to the N-terminal portion of the polypeptides, where the J-domain is located (Figure 1).

In addition, we observed a lower degree of conservation in the other protein domains and especially (32% identity) in the linker-CDTI region (Figure 1). Importantly, we noticed a low degree of conservation of amino acids previously reported to be crucial for *E. coli* DNA-binding activity (indicated by grey and black arrowheads in Figure 1 [8,11]). Finally, of interest is the observation that of the two residues, W287 and L290, crucial for *E. coli* CbpA dimerization [10], only one is conserved in the *H. pylori* CbpA protein (Figure 1, empty arrowheads).

### 3.2. H. pylori CbpA Displays a DnaK Co-Chaperone Activity

In order to start a functional characterization of *H. pylori* CbpA, we decided to explore the possibility that this stress-responsive protein can function as a DnaK co-chaperone. To this end, we expressed and purified the recombinant CbpA protein and all the members of the DnaK chaperone system of *H. pylori*, namely the DnaK chaperone, the DnaJ co-chaperone and the nucleotide exchange factor GrpE. All these proteins were expressed in *E. coli* BL21(DE3) and purified through Ni^2+^-nitrilotriacetic acid (NTA) affinity chromatography (Figure 2A).

Then, we used the purified proteins to carry out a colorimetric in vitro assay measuring the ATPase activity of DnaK in the absence or presence of the DnaJ co-chaperone and of the CbpA protein. As reported in Figure 2B, in the absence of DnaJ, DnaK possesses a weak ATPase activity, which is significantly stimulated following the addition of its co-chaperone DnaJ. Interestingly, we observed a further stimulation of the ATPase activity of DnaK when we substituted DnaJ with the CbpA purified protein. To exclude the possibility of a contamination of protein preparations by ATPases, CbpA and DnaJ purified proteins were also analysed alone for their ATPase activities (results are reported in the Appendix A). Together with the observation that *H. pylori* CbpA has a highly conserved N-terminal J-domain (Figure 1), these data clearly indicate that this protein is able to stimulate DnaK ATPase activity, thereby representing an additional member of the *H. pylori* chaperones repertoire.

### 3.3. H. pylori CbpA Is a DNA-Binding Protein

According to sequence comparison shown in Figure 1, *H. pylori* CbpA has a well-conserved N-terminal J-domain followed by a less conserved linker-CDTI region, with several substitutions in amino acids previously described as crucial residues in *E. coli* for CbpA DNA-binding activity. To explore the possibility that, regardless these mutations, *H. pylori* CbpA possesses DNA-binding activity, we carried out an electrophoretic mobility shift assay (EMSA) in which a plasmid DNA probe harboring *H. pylori* genomic DNA was incubated with increasing amounts of purified CbpA and then separated through an agarose gel. We decided to use a plasmid DNA probe in analogy with what has been previously done to characterize *E. coli* CbpA DNA binding activity [8,10,11]. The result of the EMSA assay is shown in Figure 3.

Upon addition of CbpA to the reaction, a smear of DNA is clearly detected in a dose-dependent manner (Figure 3, lanes 3 to 10). This behavior is in line with what has been previously observed for *E. coli* CbpA-DNA complexes [11]. Thus, the *H. pylori* CbpA binds and retards both the supercoiled as well as the relaxed circular isoforms of plasmid DNA probe. CbpA DNA-binding is barely detectable at 110 nM CbpA (Figure 3, lane 3), while probe saturation is observed at 340 nM CbpA (Figure 3, lane 9). Therefore, regardless of sequence divergences between *E. coli* and *H. pylori* CbpA, the *H. pylori* CbpA protein also has the capacity to bind DNA.

### 3.4. H. pylori CbpA Is a Dimer in Solution and Dimerization Is a Prerequisite for DNA-Binding

The sequence alignment reported in Figure 1 highlights the conservation of several residues between *E. coli* and *H. pylori* CbpA also in the C-terminal domain of the proteins (CTDII domain). It has previously been shown that dimerization of *E. coli* CbpA is mediated by a hydrophobic surface, comprising amino acid W287 and L290, located close to the C-terminus of the protein [10]. Considering that *H. pylori* CbpA sequence has the hydrophobic residue W substituted by the polar amino acid E (Figure 1), we decided to investigate the oligomeric state of *H. pylori* CbpA by an in vitro crosslinking assay. To this aim, the purified protein was treated with increasing concentrations of the glutaraldehyde cross-linking agent and analyzed through SDS-PAGE. From this analysis, following glutaraldehyde treatment we detected one band with apparent molecular mass just below 70 kDa (Figure 4A, left panel), the intensity of which increased in parallel with increasing amounts of cross-linking agent. By contrast, in the untreated sample we observed the monomer only (33 kDa) and no higher order oligomer of the protein (Figure 4A, left panel, lane 0). Very likely, the band observed upon protein cross-link represents the dimeric form of the protein, suggesting that *H. pylori* CbpA forms a dimer in solution. Then, to assess if *H. pylori* CbpA dimerization is driven by amino acids located close to the C-terminus of the protein, including the conserved terminal W288, we expressed and purified a mutant isoform of CbpA, in which the last 12 amino acids of the protein were deleted (CbpA-ΔCt). Then, we investigated the ability of CbpA-ΔCt to form dimers in solution through in vitro crosslinking. From the SDS-PAGE analysis reported in Figure 4A (right panel), we were able to observe the band corresponding to the monomeric form of the protein, while the higher molecular weight band of the dimer observed for CbpA-wt was absent even at the higher glutaraldehyde concentration. Similar results were obtained when we carried out in vitro crosslinking assays using a different homobifunctional crosslinker (Appendix A). These results suggest that the deleted C-terminal portion of CbpA harbors crucial residues for protein dimerization.

To investigate if CbpA dimerization is necessary for DNA recognition and binding, we carried out EMSA experiments on a plasmid DNA probe, comparing the deletion mutant CbpA-ΔCt with the wild-type protein. As reported in Figure 4B, CbpA-ΔCt showed a marked decrease in its DNA-binding capacity, if compared to the wild-type protein. Specifically, while 300 nM CbpA-wt is sufficient to bind and retard the entire amount of the DNA probe (Figure 4B, lane 4), a merely detectable retardation effect is observed at the same concentration of CbpA-ΔCt (Figure 4B, lane 9). Thus, we conclude that CbpA dimerization is a prerequisite for DNA recognition and binding.

### 3.5. Preliminary Evidence of CbpA DNA-Binding Modulation by the HspR Repressor

We have previously demonstrated that CbpA directly interacts with the HspR repressor, the master transcriptional regulator of heat-shock genes in *H. pylori*. Furthermore, we showed that CbpA modulates HspR binding to DNA, thereby acting as an important modulator of HspR-mediated transcriptional control of chaperones encoding genes [4]. Considering this CbpA-HspR functional interplay, we hypothesized that in turn HspR could affect CbpA binding to DNA. On the other hand, we cannot rule out the possibility that the interaction of CbpA with HspR might have no effects on CbpA DNA-binding. To gain novel insights on this point, we carried out a competitive EMSA assay, in which we monitored CbpA binding to DNA alone or in combination with HspR repressor. As reported in Figure 5, it appears that the inclusion of HspR in the binding reaction negatively modulates CbpA binding to DNA.

In detail, the negative effect of HspR on CbpA binding can be appreciated by comparing lanes 3, 4 and 5 with lanes 8, 9 and 10 of Figure 5. In fact, while upon the addition of increasing concentrations of CbpA alone to the reaction, a smear of DNA is clearly detected in a dose-dependent manner (Figure 5, lanes 3 to 5), in the presence of the same concentrations of CbpA incubated with HspR both plasmid isoforms are still detectable and DNA smearing is much less pronounced (Figure 5, lanes 8 to 10). Possibly, the presence of HspR in the reaction titrates out CbpA from DNA-binding. Thus, this preliminary observation supports the hypothesis that, besides its roles as a DnaK co-chaperone and as a DNA-binding protein, CbpA could modulate HspR DNA-binding activity.

## 4. Discussion

The widespread gastric pathogen *H. pylori* controls the stress-responsive expression of its major chaperone proteins through a complex strategy involving transcriptional and posttranscriptional mechanisms of regulation [2]. In this context, we have previously demonstrated that the protein encoded by the HPG27_RS02130 gene, named CbpA for its sequence similarity to the well-characterized protein of *E. coli*, takes part in heat-shock genes’ regulation by negatively modulating the DNA-binding activity of the master heat-shock repressor HspR [4]. In this work, we focussed our efforts on the characterization of CbpA activity. Starting from the analysis of sequence similarity between *H. pylori* and *E. coli* CbpA proteins, we demonstrated for the first time that *H. pylori* CbpA has a DnaK co-chaperone activity in vitro, stimulating DnaK-mediated ATPase hydrolysis in the presence of the nucleotide exchange factor GrpE (Figure 2, panel B). In addition, we showed that *H. pylori* CbpA is a dimeric DNA-binding protein (Figure 3 and Figure 4) and that this activity is negatively modulated by the heat-shock master repressor HspR (Figure 5).

*H. pylori* G27 genome harbours a gene, named HPG27_RS06700, which encodes an additional DnaK co-chaperone DnaJ, a protein displaying all the sequence features (J-domain, DnaJ C terminal domain and the DnaJ central cysteine-rich domain) typical of DnaJ/Hsp40 proteins. Of note is the observation that the HPG27_RS06700 gene does not belong to the multicistronic heat-shock operons, whose transcription is responsive to changing environmental conditions. In addition, HPG27_RS06700 transcription appears not affected by *hrcA* or *hspR* inactivation as well as by heat-shock [22,23]. On the other hand, CbpA is encoded by the leading gene of the *hspR* operon and its transcription is responsive to heat-shock and other stress insults [24]. Moreover, *cbpA* is transcriptionally controlled by the master repressor HspR, which keeps CbpA expression to a basal level until the perception of a stress signal [3,25]. Considering these observations and the DnaK co-chaperone function of CbpA, demonstrated for the first time in the present work, it can be speculated that *H. pylori* employs CbpA as the principal co-chaperone during heat-shock/stress response. The DnaJ protein encoded by the HPG27_RS06700 gene, instead, could have a different DnaK co-chaperone role and could be employed by the bacterium to respond to different stress or mainly during normal growth conditions.

Firstly isolated for its tendency to bind intrinsically curved DNA molecules, *E. coli* CbpA DNA-binding activity has been extensively characterized during the last two decades. Nowadays we know that upon dimer formation, *E. coli* CbpA binds AT-rich genomic sequences by recognizing the DNA minor groove. It has been shown that several residues mapping in the linker-CTDI protein domains are important for DNA binding, with R116 playing a predominant role. Following DNA recognition, *E. coli* CbpA protects cellular DNA from damage by forming protein-DNA aggregates, which appear similar to those formed by the nucleoid-associated protein Dps [10,26]. Noticeably, despite the lack of conservation of amino acids previously described as crucial residues in *E. coli* for CbpA DNA-binding activity (Figure 1), we show in this work that *H. pylori* CbpA has the capacity to bind a probe containing a *H. pylori* genomic DNA fragment (Figure 3). Moreover, we demonstrate that CbpA in solution is able to dimerize and that dimerization is necessary for DNA binding (Figure 4). Considering that *H. pylori cbpA* is up-regulated following exposure of cells to a variety of stress insults [24], an intriguing possibility is that CbpA could play a DNA protective role, by contacting several genomic binding sites and spreading throughout the bacterial chromosome, in analogy to what has been observed in the model organism *E. coli* [11]. Based on functional similarities between the *E. coli* and *H. pylori* CbpA proteins and considering that *H. pylori* displays an AT-rich genome, it would be not surprising if in *H. pylori* also this protein binds to AT-rich genomic sequences.

One of the most intriguing aspects of CbpA function is its interplay with the heat-shock regulator HspR, shown schematically in Figure 6. As already mentioned above, it is well established that, upon direct interaction, CbpA can modulate HspR DNA-binding capabilities, which impacts on HspR regulatory activity [4]. In this work, we show preliminary evidence supporting the possibility that this protein–protein regulation could affect also CbpA activities. Specifically, we reported a preliminary observation suggesting that HspR can negatively modulate at least CbpA DNA-binding activity (Figure 5), while further efforts are needed to assess whether HspR influences also CbpA co-chaperone function. In other words, our data suggest that HspR can be regarded as a CbpA modulator. In *E. coli*, the gene lying immediately downstream *cbpA* codes for a protein modulator of CbpA activities called CbpM [12,13]. Interestingly, the crystal structure of CbpM reveals a strong similarity to the members of the MerR family of transcription regulators, to which HspR belongs [27]. The region of the highest similarity between CbpM and MerR-like regulators is represented by the N-terminal portion, where is confined the DNA-binding domain of the transcriptional regulators [13].

In conclusion, data presented in this work support the idea of a reciprocal functional modulation between CbpA and HspR, an interaction that links HspR-mediated heat-shock gene regulation with CbpA-dependent protein folding and nucleoid maintenance, and stimulates further studies to better characterize this interplay.

## Figures and Tables

**Figure 1 microorganisms-08-00251-f001:**
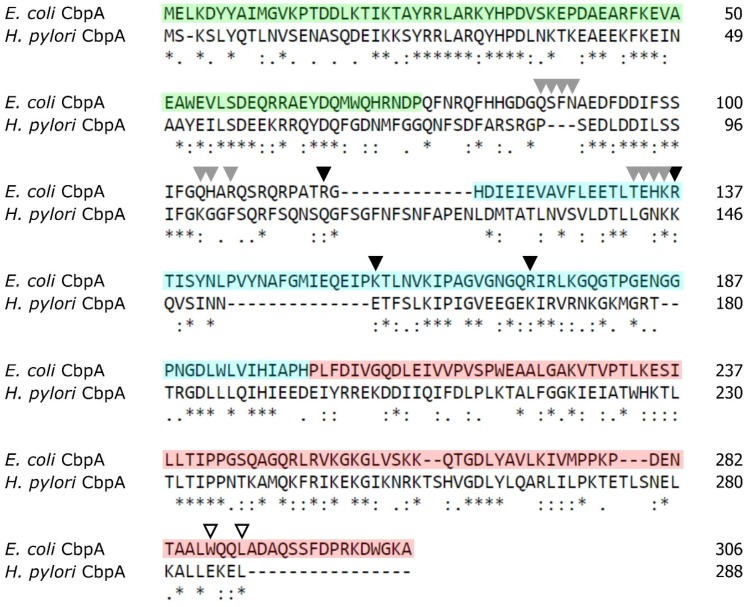
Conservation between *E. coli* and *H. pylori* CbpA proteins. Primary sequence comparison between *E. coli* and *H. pylori* CbpA was carried out using the T-Coffee Multiple Sequence Analysis online tool [21]. Symbols: asterisk (*), amino acid identity; colon (:), strong amino acid conservation; period (.), weak amino acid conservation [21]. The experimentally characterized functional domains of *E. coli* CbpA are indicated as follows: J-domain (aa 1 to aa 75): green; linker (aa 76 to 117): not highlighted; CTDI (aa 118 to aa 201): cyan; CTDII (aa 202 to aa 306): red. Grey and black arrowheads indicate residues involved in DNA binding (according [8] and [11], respectively), while empty arrowheads denote crucial residues for CbpA dimerization (according to [10]).

**Figure 2 microorganisms-08-00251-f002:**
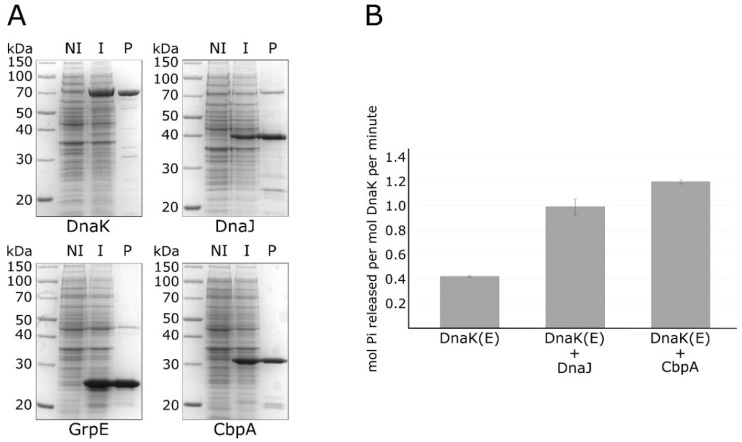
Purification of recombinant chaperone proteins and in vitro ATPase assay. Panel (**A**) sodium dodecyl sulfate–polyacrylamide gel electrophoresis (SDS-PAGE) showing the DnaK, DnaJ, GrpE and CbpA purified proteins (marked underneath each panel) following Ni–NTA affinity chromatography. Symbols: NI, total protein extract of non-induced bacteria; I, total protein extract from induced bacteria; P, purified protein. Panel (**B**) ATPase activity of DnaK measured alone (DnaK (E)), with DnaJ (DnaK (E) + DnaJ) or with CbpA (DnaK (E) + CbpA). Standard error was calculated from 3 independent experiments. The (E) symbol indicates that all the reactions included the nucleotide exchange factor GrpE.

**Figure 3 microorganisms-08-00251-f003:**
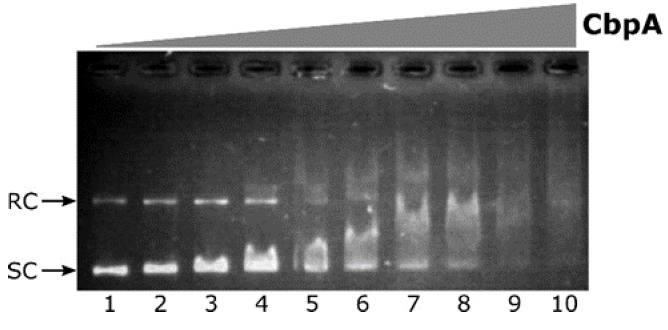
Electrophoretic mobility shift assay (EMSA) experiment with purified *H. pylori* CbpA on a plasmid DNA probe. The pHel2-flaAfur plasmid probe (7 fmol/rxn) was incubated with 0, 75, 110, 150, 190, 225, 265, 300, 340 and 375 nM of purified CbpA (lanes **1** to **10**, respectively) at 25 °C for 25 minutes before electrophoretic separation through 0.9% agarose gel. Symbols: SC, supercoiled plasmid DNA; RC, Relaxed Circular plasmid DNA.

**Figure 4 microorganisms-08-00251-f004:**
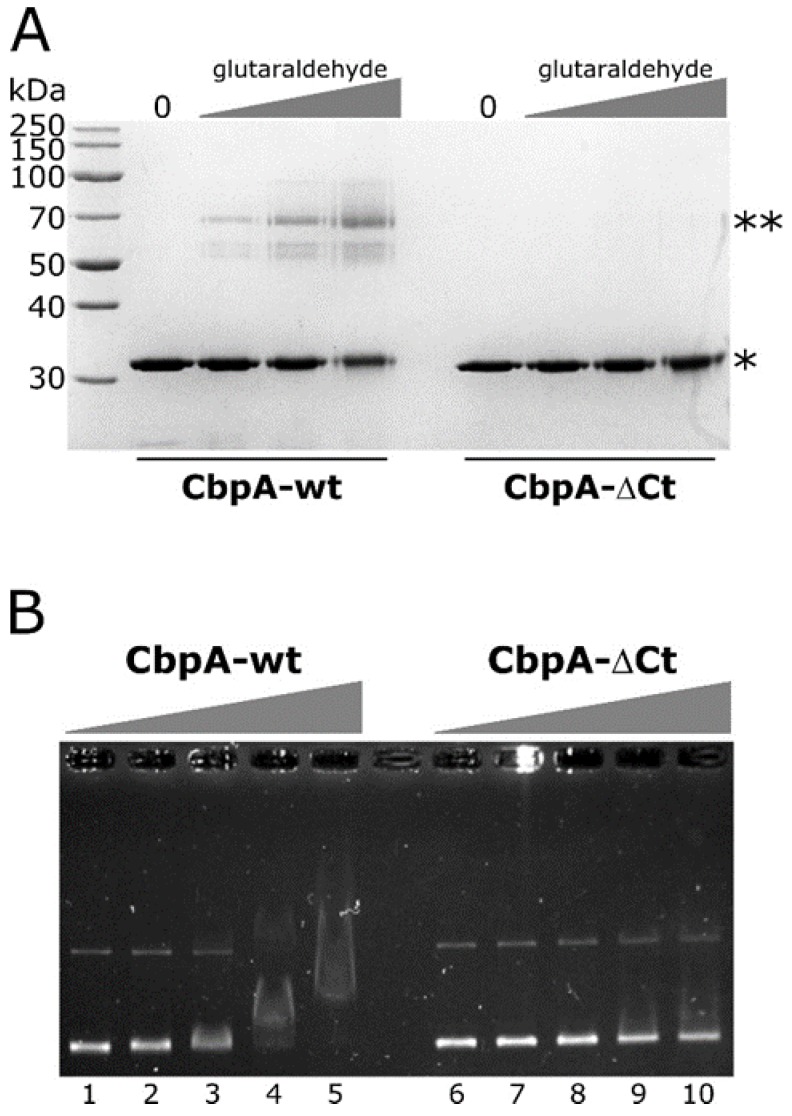
In vitro crosslinking and EMSA assays of CbpA-wt and CbpA-ΔCt proteins. Panel (**A**) SDS-PAGE of the purified CbpA-wt and CbpA-ΔCt proteins not treated (lane 0) or treated with increasing concentrations of glutaraldehyde. Symbols: *, band of the monomeric form of the proteins; **, band of the putative dimeric form of the CbpA-wt protein. Panel (**B**) EMSA assay with purified *H. pylori* CbpA-wt and CbpA-ΔCt proteins. The pHel2-flaAfur plasmid was incubated with 0, 75, 150, 300 and 600 nM of purified CbpA-wt (lanes **1**–**5**) or CbpA-ΔCt (lanes **6**–**10**) at 25 °C for 25 minutes before electrophoretic separation through 0.9% agarose gel.

**Figure 5 microorganisms-08-00251-f005:**
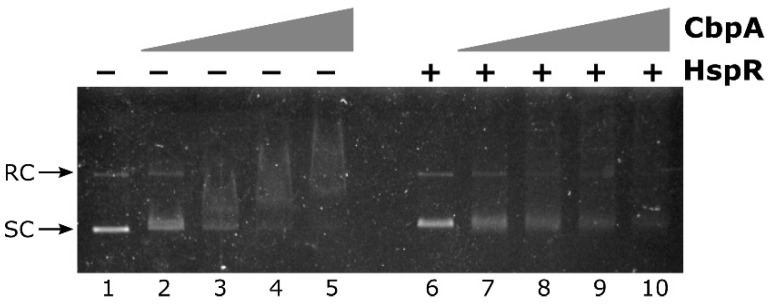
CbpA-HspR competitive EMSA assay. The pHel2-PflaAfur plasmid DNA probe was mixed with 0, 150, 225, 300 and 375 nM of CbpA in the presence of 300 nM of HspR (lanes **6**–**10**) or bovine serum albumin (BSA) as control protein (lanes **1**–**5**), incubated at 25 °C for 25 min before electrophoretic separation through 0.9% agarose gel. Symbols are as in Figure 3.

**Figure 6 microorganisms-08-00251-f006:**
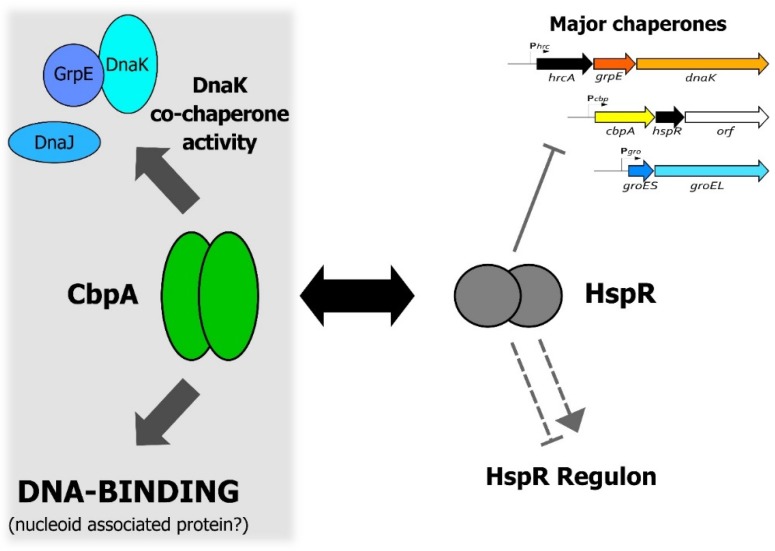
A model representing the functional interplay between CbpA and HspR in *H. pylori*. The figure shows a schematic representation of both CbpA and HspR functional activities and highlights the CbpA-HspR direct protein–protein interaction (black double arrow). Symbols are as follows: the grey filled hammerhead indicates HspR direct repression of the multicistronic operons containing the major *H. pylori* chaperone genes; the grey dashed hammerhead and arrowhead depict negative and positive indirect HspR-mediated regulation of target genes, respectively [23]. Confirmed activities of CbpA are depicted on the right-hand part of the figure, while more speculative activities are confined to the left-hand part of the figure on the grey background.

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
