# Peer review of "The *Helicobacter pylori* HspR-Modulator CbpA Is a Multifunctional Heat-Shock Protein"

_microorganisms, 2020, doi:10.3390/microorganisms8020251_

Round 1

Reviewer 1 Report

The work entitled “The Helicobacter pylori HspR-modulator CbpA is a multifunctional heat-shock protein” presents preliminary data concerning the role of CbpA protein in H. pylori response to stress. CbpA is homologous to E. coli CbpA (Co-chaperone-curved DNA binding protein A). In previous work of the same group, CbpA was shown to interact with the heat-shock master repressor HspR (doi:10.1128/JB.05295-11), preclude HspR interactions with DNA, and to be involved in timely shut-off response to stress mediated by HspR.

In this communication, the authors continue their studies on CbpA and conclude that: 1/CbpA binds DNA thus it may play a similar role in DNA protection as some nucleoid-associated proteins (e.g. Dps); 2/dimerisation is required for CbpA-DNA interactions; the authors defined the region of the protein required for dimerization; 3/HspR inhibits CbpA-DNA interactions. 4/ CbpA stimulates DnaK ATPase activity; thus, it may be involved in DnaK chaperone activity.

The overall conclusion of these and previously published results is that CbpA may provide crosstalk between chaperone DnaK and HspR stress response.

In general, the work is interesting and adds a new “element” to the H. .pylori stress response mechanism. My major concern is about the conclusions which are drawn on the basis of experiments performed using only one technique; some, in my opinion, lack proper controls. The authors point out that some data are preliminary; however, even the preliminary data should be very solid and reliable when individual experiments are concerned, while they may be speculative when the whole mechanism is discussed.

Therefore I suggest to add some more data confirming the results and make them strong, the preliminary basis for further studies.

In particular:

1/ Fig. 2, Stimulation of DnaK ATPase activity by CbpA. In my opinion, this experiment lacks control in which CbpA and DnaJ were analysed for their ATPase activities. Even if these proteins lack ATPase activities (which is the case), the purified recombinant protein fractions might be contaminated by ATPases (see Fig. 2A, there are additional bands in “P” lanes). Adding these two controls should make the results stronger. It would be very good if the authors can show a direct interaction between DnaK and CbpA.

2/ Fig. 4A Dimerisation between CbpA is shown by one method only (crosslinking). Glutaraldehyde crosslinking is sometimes misleading. The CbpADCt, in a way, constitutes a control reaction for CbpA dimerisation, but it would be ideal if the authors could show this dimerization by another method to make the results stronger. In addition, the authors state that the purified CbpA proteins were stored ain 1x Binding buffer which contains 10 mM Tris. Tris quenches glutaraldehyde and is often used to stop the reaction. Were CbpA proteins highly diluted before crosslinking or the buffer was exchanged? What was the reaction buffer used in crosslinking?

3/ Fig. 5 HspR inhibition of CbpA-DNA interactions. This is the weakest part of the manuscript. The gels are of poor technical quality. The CbpA binding to SC form of the plasmid is not really convincing (only slight smear in lane 5 on the left-hand panel), the binding to RC form is only a little bit better – the band disappears, but due to the uneven grey-scale of the gel, it is not really convincing. There is no real inhibition of the CbpA binding by HspR to the SC form of the plasmid (lane 5 on the right-hand panel); the inhibition to RC form is a little bit more obvious. Nonetheless, these results should be improved. I suggest to repeat the experiment, use a higher concentration of CbpA (on Fig. 3 up to 7.5 pmol of CbpA is used) to make the binding more visible and convincing, and then analyse the influence on HspR on CbpA-DNA interactions. Maybe the use of a linear DNA instead of a supercoiled plasmid will help? Please also correct statement in lane 329 “if these observations will be confirmed”. Even if the authors present preliminary data, these need to be confirmed as partial data, i.e., the inhibition of CbpA-DNA interactions by HspR has to be convincing and specific. Otherwise, the authors question their own results.

Minor comments:

Lane 87: “digested and cloned NdeI/XhoI in the pET22b vector” – please correct the sentence, cloned into NdeI/XhoI sites?

Lane 126, please specify how much protein was used in crosslinking experiments

Lane 177: please explain that DnaK(E) states for DnaK+GrpE

Lane 192: please specify why you use a plasmid DNA not a linear plasmid or a PCR fragment

Lane 316: is it important that the plasmid contains H. pylori genomic DNA or it is not crucial for the story. Can the authors speculate whether CbpA binding to DNA is in anyway specific (as E. coli CbpA is specific towards AT-rich regions)

Figure 6: please specify and depict it on the figure, which activities o CbpA were confirmed (generally right-hand part) and which are more speculative (left –hand part).

Please use the same specification of the protein amount through the manuscript, either the amount (pmol) or concentration (pM) (e.g. Fig 3 the CbpA amount is presented as pmol (lanes 200-201), while in the manuscript part the concentration is given (lanes 208-209))

Author Response

an item-by-item response is provided in the attached file.

Reviewer 2 Report

The paper “The Helicobacter pylori HspR -modulatori CbpA…” by Pepe et al., is focused on the functional characterization of CbpA, a regulatory protein involved in the modulation of the heat shock response in H.pylori. In particular, considering the homology with the E.coli CbpA, the authors have asked whether also in H.pylori CbpA was able to form dimers and to bind DNA in order to protect DNA damage during stress condition.

Data obtained through EMSA assays and crosslinking experiments clearly indicate that dimerization is a prerequisite for DNA recognition and binding and that the deletion of the C‑terminal portion of CbpA severely affects capacity to form dimers. As for the interaction of CbpA with other proteins, the authors were able to convincingly show that the protein is able to stimulate the ATPase activity of the DnaK chaperone, and that in vitro its DNA binding activity is decreased in the presence of HspR, the heat shock master repressor

The paper is very well written and is supported by a logical and well thought experimental approach. The results are clear and well discussed. Future perspectives addressing the multifunctional role of CbpA and its interactions with other regulatory elements are appropriately presented.

Overall, the paper gives a sound contribution to the deciphering the complex regulatory strategies adopted by H.pylori and therefore I strongly recommend the publication in Microorganisms.

Author Response

We thank the reviewer.

Round 2

Reviewer 1 Report

Thank you for making all the requested corrections, the manuscript has improved and now better presents results.